# A Hospital Medical Record Quality Scoring Tool (MeReQ): Development, Validation, and Results of a Pilot Study

**DOI:** 10.3390/healthcare12030331

**Published:** 2024-01-27

**Authors:** Alessandra Torsello, Mariarosaria Aromatario, Matteo Scopetti, Lavinia Bianco, Stefania Oliva, Stefano D’Errico, Christian Napoli

**Affiliations:** 1Department of Medical Surgical Sciences and Translational Medicine, “Sapienza” University of Rome, 00189 Rome, Italy; matteo.scopetti@uniroma1.it (M.S.); christian.napoli@uniroma1.it (C.N.); 2Sant’Andrea Hospital, University La Sapienza, 00189 Rome, Italy; maromatario@ospedalesantandrea.it; 3Department of Public Health and Infectious Diseases, “Sapienza” University of Rome, Piazzale Aldo Moro 5, 00185 Rome, Italy; lavinia.bianco@uniroma1.it (L.B.); stefania.oliva@uniroma1.it (S.O.); 4Department of Medicine, Surgery and Health, University of Trieste, 34137 Trieste, Italy; sderrico@units.it

**Keywords:** hospital medical record quality, clinical risk management, quality index, quality score, patient safety

## Abstract

Hospital medical records are valuable and cost-effective documents for assessing the quality of healthcare provided to patients by a healthcare facility during hospitalization. However, there is a lack of internationally validated tools that measure the quality of the whole hospital medical record in terms of both form and content. In this study, we developed and validated a tool, named MeReQ (medical record quality) tool, which quantifies the quality of the hospital medical record and enables statistical modeling using the data obtained. The tool was applied to evaluate a sample of hospital individual patient medical records from a secondary referral hospital and to identify the departments that require quality improvement interventions and the effects of improvement actions already implemented.

## 1. Introduction

Hospital medical records, which are vital for evaluating and enhancing the quality of care, are a useful instrument for assessing clinical risks in the context of growing complexity in healthcare services [1,2,3,4,5].

Hospital medical record review is a vital process in medical malpractice lawsuits [6,7,8] and is a fundamental document that reports the events that occurred during the patient’s hospitalization, including the drugs administered, the interventions performed, and the patient’s health status on a daily basis, at the time of hospitalization and discharge. It is crucial to have precise documentation of the medical procedures performed on the patient, the information conveyed to them, and the time and location of the procedures in case of a legal dispute, also considering that hospital medical records are used in courts of law [9]. Zhou et al. found that low-quality hospital medical records were a major factor for unfavorable verdicts in cases of severe brain injury during childbirth [10]. Similarly, Domingues et al. reported that over 80% of obstetrics lawsuits in Portugal involved poor-quality medical records [11].

Among various risk management strategies, reviewing the quality of completeness, appropriateness, and comprehensiveness of patients’ clinical records is a highly effective method to detect medical malpractice [12,13]. For instance, Hincker et al. conducted an analysis of perioperative medical documentation in a Canadian hospital and found that frequently antibiotic prophylaxis was administered at inappropriate times [14]. The hospital medical record is a multifunctional and accessible instrument for evaluating the quality of healthcare. However, there are no standardized and validated criteria to measure the overall quality of the hospital medical record. Existing scores focus on surgical or anesthetic activities, but they are either too rigid or too specific to the healthcare facility where they were developed. For instance, Curtis et al. [15] developed a score for anesthesiologic documentation based on the standards of the Royal College of Anaesthetists and the Australian and New Zealand College of Anaesthetists. Nyamulani et al. modified the score of the Royal College of Surgeons of England to assess the documentation in the operating room in 2018 [16]. The STAR score, suggested by Tuffaha et al. [17], is another instrument for surgical documentation that is not applicable to the entire medical record [18]. Hung et al.’s study focuses exclusively on the quality of notes to be written by medical residents [19].

There is a lack of validated methods in the main scientific literature databases for assessing the quality of a complete hospital medical record. Hence, a tool that can measure the quality of the entire hospital medical record is required. Specifically, healthcare professionals require independent tools that can be utilized by anyone, regardless of their specialization in quality assurance. Quality verification is not restricted to the moment of accreditation or routine checks by specialized risk management personnel. Instead, using an independent evaluation system can be a valuable exercise to cultivate a culture of quality care among all healthcare facility staff.

This study aims to create a scoring tool, named the medical record quality (MeReQ) tool, which can assess the hospital medical record’s formal and substantive aspects.

The aim was to design a tool that can achieve the following:Enables easy and reliable assessment of hospital medical record quality;Applies across various medical specialties and settings;Identifies the specific sections of the hospital medical record that require improvement;Assesses the effectiveness of enhancement measures;Facilitates comparability of quality across different departments or over different time periods.

## 2. Materials and Methods

This research employed the following methodology: developing a tool, validating the tool, and applying the tool. This study took place at a secondary referral hospital in Rome (Central Italy) that has about 450 beds and provides around 1,300,000 services annually for both inpatients and outpatients.

### 2.1. Tool Design

A draft checklist was developed by adapting the tool proposed by Pomara et al. [20]. The elements that could be assessed in the hospital medical records of each department and hospital were selected. The selected items were then categorized into five different sections:Completeness: Availability of each component and each form of the medical record;Operative procedure: Availability of documents related to the invasive procedures;Accuracy: Completion of medical and nursing diaries;Tracking: Legible signatures and identification numbers of health professionals;Informed consent: Quality of the informed consent forms administered.

### 2.2. Validation Process

The draft checklist was subjected to a face validity assessment by a panel of experts (10) who had expertise in epidemiology, healthcare management, and legal medicine. The panel was instructed to evaluate the selected items and to propose modifications for any unclear, redundant, or missing items. After this initial validation step, a cross-sectional study was conducted to confirm the five sections identified nationally and to assign a score to each section. According to the Italian law n.128 enacted on 27 March 1969, the Medical Director is in charge of the supervision and storage of the hospital medical records. The Medical Director is a medical doctor with a specialization in one of the disciplines of public health or in an equivalent discipline; alternatively, they must have carried out technical healthcare management activities in public or private healthcare institutions for at least five years Therefore, an online survey among the Medical Directors of each of the 99 Italian public local health units (PLHU) was conducted. We obtained the list of all the PLHU from the official website of the Italian Health Ministry (https://www.salute.gov.it/portale/documentazione/p6_2_8_1_1.jsp?id=13link, accessed on 5 October 2022). The enrolled Medical Directors were asked to evaluate the draft checklist by reporting any missing items and assigning a maximum score on a five-point Likert scale to each section of the hospital medical record. All the selected sections received the same maximum score, indicating that the interviewees gave equal importance to the investigated sections. A final score of 2 was attributed to each item (0, if the item was absent or blank; 1, if the item was only partially completed; 2, if the item was correctly completed and had a specific assessment score, if applicable) and to the whole hospital medical record, as the sum of the scores of each item divided by the number of items that could be evaluated in the single medical record under evaluation. In this way, the total score, being an arithmetic mean of the individual item scores, could range from 0 to 2. The original questionnaire was also modified: in addition to the questions about the importance of the different sections of the hospital medical records, some "filler" questions with grammatical and semantic errors were added to ensure answer variability and test the understanding of the original questions (intelligibility).

### 2.3. Pilot Study Methods

The tool’s effectiveness and usability were assessed. In 2021, the hospital where the study was settled comprised 25 distinct departments. However, in 2022, an additional department was established, thereby increasing the total number of departments to 26. The risk management personnel at this hospital conduct regular quality checks on hospital medical records and have privacy and security clearance to access the complete archive of hospital medical records. Upon admission, patients provide their consent regarding the use of data contained in their medical records for scientific research purposes. To evaluate the quality of medical records in the hospital using the MeReQ tool, the staff responsible for verifying the quality of hospital medical records then randomly sampled 10 records from each of the 25 wards for 2021 and each of the 26 wards for 2022, resulting in a total of 510 records. The MeReQ tool was applied to assess the quality of completeness, appropriateness, and comprehensiveness of the hospital medical records for each year and to identify the factors associated with lower quality. Before initiating the quality analysis, a consensus was established regarding when to assign a score of 1 (“partially complete”) to the items. Descriptive statistics were applied to summarize the data and report the results as frequency (percentages) and mean (±standard deviation).

To demonstrate the effectiveness of MeReQ as a hospital medical record quality indicator, the data obtained using MeReQ tool were analyzed using *T*-test and $\chi^{2}$ test to achieve the following:Evaluate the impact of improvement measures;Compare the quality of the activities between different departments;Assess changes in the quality of hospital medical records over time.

### 2.4. Statistical Analysis

All the statistical analyses were performed using R-studio 4.2.2.

## 3. Results

### 3.1. The Final Tool and Its Scoring

A total of 51% of the Medical Directors interviewed responded to the questionnaire. After the validation process, the tool resulted in a final version of 35 items, distributed across five sections, with a scoring system of 0 to 2. The 35 items and their corresponding scores are presented in Table 1. The mean scores of the different sections of the clinical records were >4 for each item and the filler items had mean scores of <1 for each item, indicating a high level of clarity and comprehensibility for the interviewee. The reliability of the tool was assessed using Cronbach’s α, which yielded values > 0.95 for each section, indicating a high level of internal consistency.

### 3.2. Pilot Study Results

#### 3.2.1. Using the MeReQ Tool to Compare Quality over Time

The pilot study evaluated the MeReQ score of 510 medical records from the archives of 2021 and 2022, with 250 and 260 records for each year, respectively.

Out of the 25 departments examined in both years of this study, nine of them exhibited a statistically significant difference between 2021 and 2022. Table 2 presents the results, which show a low standard deviation of the mean MeReQ score across different departments.

Table 3 presents a detailed analysis of the quality scores achieved for each item on the MeReQ checklist in 2021 and 2022 (aggregate data from all departments).

In 2021, in the completeness section, the items with completion rates lower than 67% are as follows: family medical history, therapy administration record, pain assessment, PU risk assessment, and VTE risk assessment. In 2022, in the completeness section, the modules with the lowest completion rates are as follows: family medical history, lifestyle history, vital sign and general inspection, pain assessment, VTE risk assessment vital sign, and general inspection.

A statistically significant difference was observed in the 2022 scores compared to the 2021 scores for history of current complaint, vital sign and general inspection, therapy administration record, and PU risk assessment.

In the operative procedures section, the items with the lowest completion percentages are the surgical safety checklist, the anesthetic chart and record, and the surgical gauze and tools tracking, both in 2021 and 2022; however, the surgical safety checklist and surgical gauze and instrument tracking values exhibited a statistically significant difference in 2022 compared to 2021, with a worsening and improving trend, respectively.

The item with the lowest completion rate in both 2021 and 2022 was MAR updating, and there was no statistically significant change observed.

In the tracking section, all items exhibit relatively high values; however, it is important to note that in both 2021 and 2022, all departments utilized the EMR. Consequently, the expected values should have been 100%.

In the informed consent section, in 2021, the items with the lowest completion percentages are treatment plan, potential risk, and alternatives.

In 2022, the item treatment plan demonstrated a statistically significant improvement compared to 2021. Conversely, the items potential risk and alternatives exhibited no statistically significant changes.

The data from the medical and surgical departments for both the 2021 and 2022 samples were aggregated. Upon comparing the MeReQ scores obtained by the medical departments, no statistically significant differences were found between 2021 and 2022. However, a statistically significant difference was observed when comparing the samples of hospital medical records from the surgical departments.

#### 3.2.2. Using the MeReQ Tool to Compare Quality across Different Departments

In 2021, the mean MeReQ score of all the hospital medical records from surgical departments was 1.60; the mean MeReQ score of all the hospital medical records from medical departments was 1.45; the *T*-test shows a significant difference between these two means.

In 2022, the mean MeReQ score of all the hospital medical records from surgical departments was 1.61; the mean MeReQ score of all the hospital medical records from medical departments was 1.49; the *T*-test shows a significant difference between these two means.

Using the 2022 sample of hospital medical records, the section scores for completeness across all medical departments were calculated and compared with those of the surgical departments. The medical departments achieved a higher average score (*p*-value < 0.05).

#### 3.2.3. Using the MeReQ Tool to Evaluate the Impact of Improvement Actions

Throughout the study period, the risk management team implemented several measures to enhance the quality of care. These included the implementation of an electronic medical record system across all departments (2021), the digitization of fall and pressure ulcer risk assessment scales (2022), and the promotion of staff awareness regarding the utilization of high-quality informed consent forms and the surgical gauze and instrument tracking form in the operating room (2022).

*Impact of EMR*: All the tested departments use EMR for medical and nursing annotation registration; none of these department in 2021 and in 2022 obtained a full score in the accuracy and in the tracking sections (Table 3) due to the non-insertion of the full annotation register or because of the medical annotation register not being updated daily (the patient’s status is not documented in the notes, which are solely generated by the EMR management system. For instance, the system generates notes when a drug prescription is altered or laboratory tests are requested). After verification, it was noted that the absent diaries were composed within the EMR management system but were not printed and incorporated into the duplicate of the folder dispatched to the archive.

*Impact of digitization of PU risk assessment forms*: In 2021, the pressure ulcer risk assessment form was paper-based and required to be completed manually by the healthcare provider. It also needed to be updated periodically during the patient’s hospitalization. However, in 2022, the evaluation procedure underwent digitalization. As a result, the operator can now fill it in directly from the EMR management system. Consequently, there is no longer a need to update the completed form when the patient enters the department. Instead, an additional form is written each time the patient is reassessed. The percentage of the correct completion of the PU risk assessment form increased from 38% in 2021 to 78% in 2022, which is a statistically significant improvement (Table 3).

*Impact of promotion of staff awareness*: As depicted in Table 3, a statistically significant improvement in the quality of the informed consent forms employed in the hospital under investigation was observed when comparing 2022 to 2021. Furthermore, the surgical gauze and tools tracking module exhibited a substantial rise in utilization within the operating room.

### 3.3. Using MeReQ to Pinpoint Which Items Need Improvement

The surgical safety checklist is an integral component of the mandatory documentation for the operating room. It is compiled in a digital format on the operating room management system and a form is generated. The team operators are required to manually affix their signature on the printed form. Upon analyzing the surgical safety checklist item in detail in the hospital medical records sampled in this study, it is evident that, in 2021, it was present and correctly filled out in 37% of cases, absent (score of zero) in 3% of cases, and the team’s signatures were incomplete (score of one) in 60% of cases. In 2022, it was present and correctly filled out in 15% of cases, absent (score of zero) in 9% of cases, and the team’s signatures were incomplete (score of one) in 76% of cases.

## 4. Discussion

This study aimed to develop a valid and efficient tool for assessing the quality of the hospital medical records we reviewed. Moreover, the tool should pinpoint the specific areas of care that need improvement and measure the effects of the improvement interventions. Making the use of the audit checklist quicker and easier was one of the main objectives of designing the MeReQ tool. In fact, with a view to optimizing resources, especially in times of staff shortages, it is necessary to analyze a significant quantity of hospital medical records in an acceptable time; furthermore, the activity must not be excessively frustrating for the operator. Based on the data from the pilot study, the MeReQ tool, which was developed and validated, appears to exhibit the expected properties. Compared to hospital medical record auditing tools published in the literature, MeReQ is faster to use: although it analyzes the medical record in its entirety, the maximum number of entries is 35; in comparison, the STAR score includes 50 items only to test the quality of hospital medical documentation related to surgeries [17]. In a study by Azzolini et al., the checklist consisted of 48 items [21]. The CRABEL score includes 20 items but does not include analysis of the operating room and anesthetic documentation [22].

The MeReQ tool can be used to compare the quality of hospital medical documentation across different time periods, departments, or hospitals, without requiring advanced statistical skills. In fact, relatively simple statistical tests and techniques are sufficient to analyze the data obtained by applying the MeReQ tool. The tool can be applied without extensive training and is suitable for use by medical and nursing staff, as well as non-healthcare personnel who have received specific training in the proper compilation of health documentation.

The MeReQ tool is designed to generate data that can be inserted into computerized databases with a format that enables easy and rapid data extraction. For instance, an operator with basic computer skills could use a common spreadsheet or text file. Future developments of MeReQ may include the creation of a program that allows for automated statistical analysis and extraction of the data obtained.

The MeReQ tool is designed to be easily adaptable to different settings simply by modifying, adding, or removing one or more items. For instance, risk assessment forms that are mandatory may vary depending on local laws or healthcare facility policies: a suicide risk assessment tool may be mandatory only in psychiatric settings, or the risk of prolonged hospitalization may be calculated only in selected cases; if a hospital does not perform surgeries or invasive procedures, the items pertaining solely to the operating room from the MeReQ tool can be removed with ease.

The MeReQ tool does not lose precision and power by decreasing the number of items. In fact, departments that perform surgical operations generally have more complex hospital medical records and, therefore, more items of the MeReQ tool to evaluate. However, this does not translate into a greater probability of obtaining a lower score.

In the study sample, between 2021 and 2022, the improvement interventions focused on PU risk forms and informed consent forms. The departments that administer the informed consent forms showed a significant improvement in scores. On the other hand, medical departments, which were mainly affected only by the digitization of the PU risk card, did not show a significant improvement. This is despite the fact that the medical records were intrinsically less complex to compile.

Consequently, this study emphasizes the effectiveness of the MeReQ tool in assessing the impact of various interventions or factors that may affect the quality of hospital medical records, including staff changes, service organization modifications, and the occurrence of adverse events. Nevertheless, the overall MeReQ tool of a hospital medical record would not provide enough information to examine the various factors that contributed to the final score, but the separate block structure enables a more specific or thorough assessment. As an illustration, the MeReQ tool accurately reflected the effect of digitizing the risk assessment forms and adopting the EMR. For instance, in our pilot study, we expected the EMR to lead to optimal scores in the accuracy and tracking sections, but this was not always the case. We found that the accuracy and traceability of hospital medical records were below our expectations because the medical and nursing diaries were not archived properly. Although the diaries were written electronically, they were not printed and included in the folders that went to the archive. So, the quality analysis of the hospital medical records showed that healthcare workers needed training on how to use the EMR correctly, especially on how to fill out and archive the diaries. A previous systematic review also found that low IT literacy and insufficient training among healthcare workers prevented them from using EMRs effectively [23].

The MeReQ is a tool that enables the comparison of the scores achieved by different teams working in the same department. For instance, the scores of two distinct operational teams can be compared and followed by specific training activities. The pilot study showed that some operating room teams neglect to sign operating room checklists or document gauze counts.

The MeReQ scores derived from the informed consent forms also enable us to differentiate the quality of care of the various teams.

Compared to the majority of clinical documentation auditing tools published in the literature, whether validated or not, the MeReQ tool has the important characteristic of being designed purely from a medico-legal perspective. In order to create a quick-use control tool with a limited number of items, we focused on those aspects that appear most significant in determining the outcome of litigation. Furthermore, given the diffusion of the EMR, the MeReQ tool takes into account those entries that are now automatically present in a hospital medical record or contain information available in other ways. For instance, the CRABEL score involves checking which operator wrote down the medical history and physical examination at the time of admission [22]. Although it is important to know the authors of these notes, this information can be easily found by consulting the shifts of the healthcare personnel present in the room when the patient was admitted. If the EMR is used in the healthcare facility, the operator’s name automatically appears at the bottom of the admission notes.

The MeReQ has an important feature of a three-point rating system. In contrast, other health documentation audit tools only provide the option of “complete” or “incomplete” [17,18,22]. However, in clinical practice, it is common for forms or notes to be filled out incorrectly. From a medico-legal perspective, the difference between non-use and misuse is important. For instance, a low-quality informed consent form is not the same as a lost informed consent form, not archived, or never given to the patient: in the first instance, the hospital personnel can not demonstrate that the patient was correctly and completely informed; in the second instance, either the patient was never informed or the hospital staff were severely careless with important documents [24]. Secondly, the interventions required for improvement differ depending on whether the problem is unsuitable forms or the operator’s lack of awareness regarding the administration of informed consent.

Although other hospital medical record auditing systems use four-point scoring systems [19], we believe that this level of detail in the assessment is counterproductive. From a medico-legal perspective, it is imperative to intervene to improve both the forms that are particularly lacking in their completion and those that are written in an almost satisfactory manner. Therefore, the assessment tool does not need to differentiate how poorly one of the items was completed. Rating scales with four or more levels have a limitation in that it is challenging to standardize judgments between different operators who evaluate medical records. To ensure standardization, each level on a rating scale should be clearly defined. However, this can lead to slower and more complicated evaluation work.

This study has some limitations. First, the MeReQ tool is designed to assess the quality of hospital medical records structured according to the main guidelines in use in European countries or recommended by the WHO. Furthermore, the MeReQ tool was designed to evaluate the quality of hospital medical records used in inpatient departments, not in outpatient clinics. Currently, a substantial proportion of healthcare services are provided on an outpatient basis, and there is a lack of specialized tools to assess the quality of the associated documentation [25,26]. However, the MeReQ tool may not be sufficient to monitor the quality of hospital medical records in both inpatient departments and outpatient clinics. Other tools and methods may be required to ensure the comprehensive and accurate assessment of the quality of healthcare documentation.

Further work is required to explore the usability of the MeReQ tool in specialist healthcare settings. In particular, future studies will consider validating the MeReQ tool in other settings and other hospitals.

Another limitation of the MeReQ tool is that operators must be trained in its proper use, especially when assigning an “incomplete” score (value of one). It is essential to establish beforehand whether an item is considered “completed” or not. These criteria need to be shared and agreed upon by all operators responsible for verifying the quality of medical records using MeReQ. Moreover, the questionnaires can be filled out both electronically and manually, which can impact their completion due to the variability that may exist between people who complete them.

Lastly, in Italy, hospital medical records must, by law, have specific characteristics that might not be mandatory in other countries, and the tool was built with Italian legislation in mind. This might not encourage the use of our tool outside of Italy, regardless of its flexibility.

## 5. Conclusions

A standardized tool for evaluating the quality of hospital medical records was proposed, developed, and tested on a selected sample, despite some limitations. The MeReQ tool is a flexible tool that can evaluate the quality of particular aspects of a hospital’s medical records due to its modular block structure.

The use of the MeReQ tool may also facilitate the periodic comparison of the quality trends of the medical reports and the actions taken to improve them.

## Figures and Tables

**Table 1 healthcare-12-00331-t001:** MeReQ checklist.

	MeReQ Checklist	Score		
1.0	**Completeness**			
1.1	Front sheet: does it contain information about the patient’s identity, ID, personal details, provisional and final diagnosis, recovery unit, and procedures performed?	0 □	1 □	2 □
1.2	Family medical history: is there any information available regarding diseases and health conditions of the patient’s family?	0 □	1 □	2 □
1.3	Lifestyle history: are there any records of tobacco or alcohol consumption, drug usage history, nutritional status, physical activity, and other lifestyle habits?	0 □	1 □	2 □
1.4	History of current complaint: is there any documentation regarding the reason for the patient’s hospitalization, such as a chronological account of the patient’s symptoms and signs?	0 □	1 □	2 □
1.5	Past medical history: are there any records of the patient’s medical history, including illnesses, surgeries, injuries, and allergies prior to the onset of the presenting problem?	0 □	1 □	2 □
1.6	Vital sign and general inspection: are there any observations regarding the patient’s overall appearance, physique, blood pressure, heart rate, body temperature, nails, skin, hair, and muscle mass?	0 □	1 □	2 □
1.7	Physical examination (specific): are there any notes available on the evaluation of specific organ systems, such as the head and neck, thorax, abdomen, limbs, and urinary apparatus?	0 □	1 □	2 □
1.8	Therapy administration record: is it completed daily with the patient’s name or ID, medication name, dosage strength, frequency, method of delivery, date, and time of each dose administration?	0 □	1 □	2 □
1.9	Pain assessment: are daily records kept of the pain experienced by the patient, as measured by pain intensity scales or other objective measures?	0 □	1 □	2 □
1.10	Fall risk assessment: is the patient’s risk of falling evaluated on a daily basis using risk scales?	0 □	1 □	2 □
1.11	Pressure ulcer risk assessment: are there any records of the patient’s risk of developing pressure ulcers using risk scales?	0 □	1 □	2 □
1.12	Venous thromboembolism risk assessment: are there any records of the patient’s risk of developing venous thromboembolism using risk scales?	0 □	1 □	2 □
1.13	Discharge summary: is there a comprehensive summary of the care provided, diagnosis, procedures, medications, tests, problems, treatment plan, and which professional or service the patient is referred to after discharge?	0 □	1 □	2 □
1.14	Summary sheet: if the health authority deems it necessary, is the summary sheet present and accurately completed?	0 □	1 □	2 □
2.0	**Operative procedures**			
2.1	Preoperative checklist: are all preoperative criteria met before the patient is taken to the operating theater?	0 □	1 □	2 □
2.2	Surgical safety checklist: have all the sections been completed accurately and signed?	0 □	1 □	2 □
2.3	Operation report: is there an exhaustive and clear description of all the phases of the surgical procedure performed?	0 □	1 □	2 □
2.4	Postoperative checklist: are all the postoperative criteria met before bringing the patient back into the ward?	0 □	1 □	2 □
2.5	Anesthetic chart and record: are there any reported types of anesthesia, including the anesthetic administered, its type and length of administration, the time elapsed from the start of anesthesia to wake up, allergic reactions experienced, and vital signs during the procedure?	0 □	1 □	2 □
2.6	Implanted devices recording: are the labels of the implanted devices clearly visible and readable? Is the serial number along with the manufacturer information also visible and readable?	0 □	1 □	2 □
2.7	Surgical gauze and tools tracking: is there a list of all the gauze and tools used? Were they counted before and after the procedure?	0 □	1 □	2 □
3.0	**Accuracy**			
3.1	Medical annotation register updating: do the doctors update the annotations on a daily and regular basis?	0 □	1 □	2 □
3.2	Medical annotation register legibility: are the medical annotations written in a clear and legible manner?	0 □	1 □	2 □
3.3	Nursing annotation register updating: do the nurses update the annotations on a daily and regular basis?	0 □	1 □	2 □
3.4	Nursing annotation register legibility: are the nursing annotations written in a clear and legible manner?	0 □	1 □	2 □
4.0	**Tracking**			
4.1	Medical annotation register signing: is each annotation clearly signed and identified with a visible signature and ID of its author?	0 □	1 □	2 □
4.2	Nurse annotation register signing: is each annotation clearly signed and identified with a visible signature and ID of its author?	0 □	1 □	2 □
4.3	Operative record signing: did the healthcare worker who performed the procedure sign and stamp the operative record?	0 □	1 □	2 □
5.0	**Informed consent**			
5.1	Patient identification: on the consent form, are the patient’s name, ID, date of birth, and place of birth reported?	0 □	1 □	2 □
5.2	Diagnosis: is there any information about the pathology for which the procedure is being performed and/or the therapy is being administrated?	0 □	1 □	2 □
5.3	Treatment plan: does the description of the therapeutic and/or diagnostic intervention provide sufficient clarity and detail?	0 □	1 □	2 □
5.4	Healthcare provider identification: does the form have the stamp and signature of the healthcare worker (doctor, nurse, or other healthcare professional) who provided informed consent to the patient?	0 □	1 □	2 □
5.5	Patient’s signature: is the patient’s signature on the consent form? Is it clear and legible?	0 □	1 □	2 □
5.6	Potential risk: is there a complete description of the reasonably foreseeable risks or discomforts for the patient? Does the description include information on whether a risk is reversible and the probability of the risk based on existing data?	0 □	1 □	2 □
5.7	Alternatives: is there information on other relevant options for treatment for the patient’s condition?	0 □	1 □	2 □
	**Final MeReQ Score**			

**Table 2 healthcare-12-00331-t002:** The MeReQ scores for the year 2021 and the MeReQ score for the year 2022 obtained from each of the 25 departments, and the results of the *T*-test (*p*-value).

Department	MeReQ Score 2021 (Mean and sd)	MeReQ Score 2022 (Mean and sd)	*p*-Value (*T*-Test)
D 1	1.80 (0.189373)	1.37 (0.517406)	0.05725
D 2	1.57 (0.3254)	1.61 (0.224)	0.76
D 3	1.46 (0.20380)	1.56 (0.5253)	0.6208
D 4	1.66 (0.0824)	1.38 (0.216)	0.003039
D 5	1.59 (0.18111)	1.69 (0.09574)	0.3147
D 6	1.62 (0.16930)	1.53 (0.281472)	0.1257
D 7	1.70 (0.07190)	1.55 (0.100835)	0.004038
D 8	1.71 (0.068)	1.63 (0.172)	0.1874
D 9	1.39 (0.3009)	1.65 (0.14266)	0.03816
D 10	1.50 (0.06625)	1.65 (0.05561)	5.645$e^{-5}$
D 11	1.31 (0.177008)	1.38 (0.15491)	0.1556
D 12	1.78 (0.1916)	1.71 (0.1839)	0.4815
D 13	1.30 (0.2728)	1.33 (0.14757)	0.08138
D 14	1.10 (0.070986)	1.65 (0.279880)	5.739$e^{-5}$
D 15	1.80 (0.0625)	1.77 (0.24745)	0.7488
D 16	1.42 (0.3011)	1.19 (0.1523884)	0.7809
D 17	1.58 (0.15)	1.69 (0.099)	0.06831
D 18	1.70 (0.098)	1.81 (0.086)	0.01898
D 19	1.05 (0.22236)	1.46 (0.1429)	0.0002184
D 20	1.72 (0.1177)	1.45 (0.15634)	0.0004163
D 21	1.35 (0.2363)	1.58 (0.0470)	0.01556
D 22	0.62 (0.08432)	1.16 (0.2233582)	0.0004163
D 23	1.50 (0.066257)	1.42 (0.1316)	0.1556
D 24	1.62 (0.0771)	1.39 (0.143932)	0.8421
D 25	1.64 (0.13116)	1.69 (0.1687795)	0.1527

**Table 3 healthcare-12-00331-t003:** Analysis of the quality scores achieved for each item on the MeReQ tool in 2021 and 2022 (aggregate data from all departments).

**Completeness**			
	**2021**	**2022**	* **p** * **-Value** $\left(\chi^{2}\right)$
Front sheet	100%	100%	NA
Family medical history	66%	67%	0.9235
Lifestyle history	73%	67%	0.209
History of current complaint	89%	82%	0.02943
Past medical history	91%	84%	0.02633
Vital sign and general inspection	74%	63%	0.01792
Physical examination	79%	72%	0.08963
Therapy administration record	62%	77%	0.0003743
Pain assessment	56%	61%	0.2751
Fall risk assessment	73%	80%	0.1089
PU risk assessment	38%	78%	2.2$e^{-16}$
VTE risk assessment	6%	3%	0.07129
Discharge summary	91%	88%	0.3814
Summary sheet	100%	100%	NA
**Operative procedures**			
	**2021**	**2022**	* **p** * **-Value** $\left(\chi^{2}\right)$
Preoperative checklist	90%	84%	0.2288
Surgical safety checklist	37%	15%	0.0002783
Operation report	98%	100%	0.2614
Postoperative checklist	90%	96%	0.1564
Anesthetic chart and record	0%	0%	NA
Implanted devices recording	96%	100%	0.04443
Surgical gauze and tools tracking	39%	52%	0.03692
**Accuracy**			
	**2021**	**2022**	* **p** * **-Value** $\left(\chi^{2}\right)$
MAR updating	64%	59%	0.9555
MAR legibility	80%	85%	0.2087
NAR updating	93%	92%	0.7151
NAR legibility	89%	93%	0.1719
**Tracking**			
	**2021**	**2022**	* **p** * **-Value** $\left(\chi^{2}\right)$
MAR signing	83%	85%	0.7532
NAR signing	89%	92%	0.2284
OR signing	98%	100%	0.2692
**Informed consent**			
	**2021**	**2022**	* **p** * **-Value** $\left(\chi^{2}\right)$
Patient identification	90%	84%	0.1604
Diagnosis	97%	90%	0.03159
Treatment plan	46%	67%	0.0004336
Healthcare provider identification	87%	99%	0.0001861
Patient’s signature	96%	100%	0.05401
Potential risk	46%	53%	0.2322
Alternatives	20%	37%	0.002165

## Data Availability

Data are contained within the article.

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
