# Peer review of "A Hospital Medical Record Quality Scoring Tool (MeReQ): Development, Validation, and Results of a Pilot Study"

_healthcare, 2024, doi:10.3390/healthcare12030331_

Round 1

Reviewer 1 Report

Comments and Suggestions for Authors

Comments on the Quality of English Language

There are a few instances where articles (e.g. "the") are missing or inappropriately used. In a few cases, the verb tense used is present tense when it should be past tense.

Author Response

Thank you very much for your time in reviewing this manuscript. Please see the attachment with the detailed responses to your corresponding revisions/corrections. You can also find the changes highlighted  in the re-submitted manuscript.

Reviewer 2 Report

Comments and Suggestions for Authors

Report about the article Healthcare-2809955

General comment:

The study presents the development of a questionnaire to measure the quality of clinical records (from medical and surgical areas) in hospital setting. And the application of said questionnaire in a pilot study.

The questionnaire was contrasted by directors of 90 health institutions. Subsequently, the EMR assessment instrument has been validated in a pilot study carried out in two consecutive years, in a large sample of medical and surgical medical records at a 450-bed hospital in Rome.

The importance of having an instrument to measure and improve the completion of Clinical Records is evident. Furthermore, it has been described that the application of an evaluation system has, in itself, an improving effect on the procedure to which it is applied. And the results of the article confirm this idea.

A system for evaluating the quality of medical records can significantly improve medical practice and be useful for healthcare management.

The article should clarify the following points:

- The text does not state whether the study has been reviewed by an IRB or research ethics committee. Such approval is required prior to publication.

- Regarding the title (“Proposal of a national Medical Record Quality scoring tool (MeReQ): development, validation and results of a pilot study”): A change is suggested because the title refers to a national evaluation of Medical Records, but what is presented is a new questionnaire (carried out at the national level) and a pilot study.

- Regarding the Introduction: there it is stated that "there are no standardized and validated criteria to measure the overall quality of the medical record. Existing scores focus on surgical or anaesthetic activities, but they are either too rigid or too specific to the health care facility where they were developed...". However, now there are several ways of assessing EMRs through, for example, hospital assessment and accreditation systems. Although it is true that the article proposes an independent evaluation system aimed at to the evaluation of Medical Records.

- Regarding the methodology:

- The way in which the Medical Records have been accessed in the pilot study is not specified: who has accessed, with what authorization; whether or not the Medical Records were identified, if permission has been asked from patients and professionals.

- The article explains that the instrument was sent an online survey among the ‘Sanitary Directors of each of the 99 Italian public local health units (PLHU). It would be good to identify the professions of Health Directors since they can be doctors, economists, ...

It would also be good to expand the information on this consultation (response rate to surveys, etc.).

On the other hand, it would have been appropriate for hospital doctors and nurses and not just managers to review the instrument.

- It seems that the questionnaires are filled out manually, which, as the authors themselves acknowledge, conditions good completion due to the variability that may exist between people who complete it.

The homogenization and interoperability of the instrument requires the establishment of prior criteria. It is not known how this task was performed in the pilot study or how it is recommended to perform this task.

- It is explained that the preparation of the instrument has been done from a medical-legal perspective. It would be good if the authors could explain what that perspective consists of.

- In the results, it is indicated that ‘Thus, the analysis suggests that healthcare providers may not fully appreciate the medical and legal significance of filling out the forms for risk prevention Therefore, the analysis suggests that healthcare providers may not fully appreciate the medical and legal importance of completing risk prevention forms.’ It will be appreciated if you justify that result.

- If the study is going to be applied in HME, it is suggested to use an application that allows automatic dumping of the data required for the analysis.

- The measurement instrument includes several purposes: the quality of the Medical Records and also the monitoring of some prevention protocols, such as 'Fall risk assessment', 'Pressure ulcer risk assessment' or 'Venous thromboembolism risk assessment', which may or may not be implemented in other hospitals, regardless of the quality of the medical records.

- Regarding the bibliography, there are some studies carried out precisely in Italy that are not cited in the article and may be relevant, such as:

Bo M, Fiandra U, Raciti IM, Ribero S, Siliquini R, Rapellino M, Gianino MM. Quality evaluation of informed consent forms: a pilot study. Ig Sanita Pubbl. 2009 Sep-Oct; 65(5):453-65. Italian. PMID: 20010991.

Gianino MM, Raciti IM, Galzerano M, Villata E, Fonte G, Rapellino M, Fiandra U. The qualitative evaluation of the medical record: a pilot study [Evaluation of the quality of hospital medical records in a hospital in Turin (Italy)]. Ig Sanita Pubbl. 2008 Nov-Dec; 64(6):719-34. Italian. PMID: 19219084.

Author Response

(The authors gave the same response as above.)

Round 2

Reviewer 2 Report

Comments and Suggestions for Authors

I consider that the authors have answered the questions raised and introduced changes in the manuscript that allow the publication of the research.